# Assessment of the elite accessions of bael [*Aegle marmelos* (L.) Corr.] in Sri Lanka based on morphometric, organoleptic, and elemental properties of the fruits and phylogenetic relationships

Chamila Kumari Pathirana[1,2], Lahiru Thilanka Ranaweera[3], Terrence Madhujith[2,4], Kalyani Weerasinghe Ketipearachchi[5], Kumar Lakshman Gamlath[5], Janakie Prasanthika Eeswara[1,2], Suneth Sithumini Sooriyapathirana [3]*

1 Department of Crop Science, Faculty of Agriculture, University of Peradeniya, Peradeniya, Sri Lanka, 2 Postgraduate Institute of Agriculture, University of Peradeniya, Peradeniya, Sri Lanka, 3 Department of Molecular Biology and Biotechnology, Faculty of Science, University of Peradeniya, Peradeniya, Sri Lanka, 4 Department of Food Science and Technology, Faculty of Agriculture, University of Peradeniya, Peradeniya, Sri Lanka, 5 Fruit Crop Research and Development Station, Gannoruwa, Peradeniya, Sri Lanka

* sunethuop@gmail.com

**Data Availability Statement:** The DNA sequence data generated in the present study are available in

## Abstract

*Aegle marmelos* L. (Bael) is a native tree fruit species in the Indian subcontinent and Southeast Asia. Bael is a popular fruit because of its significant nutritional and medicinal properties. However, bael is an underutilized fruit species in Sri Lanka. Thus, Fruit Crop Research and Development Station of the Department of Agriculture of Sri Lanka has selected five elite bael accessions (*Beheth Beli*, *Paragammana*, *Mawanella*, *Rambukkana*, and *Polonnaruwa-Supun*). We assessed these five accessions for the variation of the fruit size, pulp, organoleptic preference, elemental properties, genetic diversity, and evolutionary history. The fruits at the golden-ripe stage were collected during the peak fruiting seasons in 2015, 2016, and 2017. The fruit size, pulp, shell thickness, and seed size were measured and subjected to the General Linear Model (GLM) and Principal Component (PC) Analyses. The fruit pulp was distributed among a group of 30 taste-panelists to rank for the parameters: external appearance, flesh color, aroma, texture, sweetness, and overall preference. The rank data were subjected to association and PC analyses. The elemental contents of the fruit pulp samples were measured using Inductively coupled plasma mass spectrometry and subjected to GLM and PC analyses. We observed a significant diversity in fruit size, organoleptic preference, and elemental contents among bael accessions. *Rambukkana* and *Polonnaruwa-Supun* yield the biggest and most preferred fruits. We used *trnH-psbA*, *atpB-rbcL* spacer, *matk-trnT* spacer, and *trnL* markers to construct phylogenies. Sri Lankan bael split from an Indian counterpart, approximately 8.52 MYA in the Pliocene epoch. However, broader germplasm of Indian bael must be assessed to see the presence of any independent evolution within Sri Lanka.

https://www.ncbi.nlm.nih.gov/nuccore/ under the accession numbers MN082783 to MN082802. The other datasets presented in this study are provided either in the paper or Supporting Information.

**Funding:** No specific funding is required.

**Competing interests:** Authors declare no completing interests.

## Introduction

*Aegle marmelos* L. (Bael) is an indigenous tree fruit species in the Indian subcontinent and Southeast Asia. Bael is a perennial crop in India, Sri Lanka, Pakistan, Bangladesh, Myanmar, Thailand, Vietnam, the Philippines, Cambodia, Malaysia, Java, and other southeastern Asian countries [1, 2]. Bael is a sacred tree in India. The gardens of many Indian Hindu temples have bael trees [3]. The ripe fruit, which contains a delicious pulp, is the most valuable part of the bael tree [4]. People mainly consume bael as fresh fruit. However, the value-added products of bael, such as drinks, traditional sweets, jam, and pudding, are available in the market [5]. People prefer bael primarily because of its rich taste and ability to cure constipation [6]. Bael fruit is an expensive commodity in supermarkets and street fruit-stalls. All parts of the bael tree possess medicinal values [7]. Thus, bael is famous as a valuable crop with immense medicinal and nutritional potentials [8]. There are many reports available on the medicinal and industrial values of bael in India [9–13]. However, bael is an underutilized crop species in Sri Lanka [14].

Bael can grow well under adverse conditions. Bael trees thrive well in high altitudes as high as 1,200 m and withstand without any significant growth retardation at extreme temperatures such as -7°C and 50°C [1]. The prolonged droughts cease fruiting in bael. However, the tree can flourish well under low moisture levels in alkaline, stony, and shallow soils. For example, the 'oolitic-limestone' soils of South Florida provide successful growth and fruit set. Fruitful bael trees dominate many vegetations in arid soils and marginal desert lands of India and Sri Lanka [1].

Bael is the solitary species in the genus *Aegle* of the subfamily Aurantioideae, which has a well-established phylogeny [15]. However, only the samples collected from India provided the basis for estimating the phylogenetic position, origin, and distribution from the natural range of bael [16, 17]. None of the published phylogenetic studies have considered the bael germplasm in Sri Lanka. Thus, the origin of bael germplasm and the genetic diversity of bael germplasm in Sri Lanka are not known.

The Fruit Crop Research and Development Station (FCRDS) of the Sri Lankan Department of Agriculture assessed the bael germplasm in Sri Lanka based on the fruit quality and the information collected from the national network of agricultural extension workers. Thereby, FCRDS identified five elite bael accessions for mass propagation using grafting and micropropagation. However, no details exist on the variation of the fruit size, pulp yield, organoleptic preference, elemental content, and genetic diversity of the elite bael accessions. Therefore, the present study assessed the morphometric variation, organoleptic desire, and elemental composition of the fruits of elite bael accessions in Sri Lanka. We also studied the phylogenetic relationships and molecular dating of the elite bael accessions in comparison to the related germplasm worldwide.

## Materials and methods

### Sample collection

We assessed five elite bael accessions of Sri Lanka in the present study. Table 1 presents the climatic and soil conditions, and Global Positioning System (GPS) coordinates of the original locations of the accessions and their names. We randomly hand-picked fruits from each accession within two-week-window of the peak fruiting seasons at golden-ripe stage (middle of the season: within February to March) of the years 2015, 2016, and 2017. The fruits were placed in cardboard boxes under ambient conditions and transported to the laboratory for measurements. We finished the data collection within 24 hours of harvesting.

**Table 1. Bael accessions, agro-climatic details and GPS coordinates of the growing locations.**

| Accession | Location, Agro-Ecological Region and District of Sri Lanka | GPS Coordinates | Mean annual rainfall (mm) | Temperature (minimum and maximum) °C | Soil type of the location |
|---|---|---|---|---|---|
| *Beheth Beli* (**BB**) | *Gannoruwa*, WM2b, Kandy | 7.277006, 80.595299 | 2083 | 18–31 (2015) | Red yellow podzolic soils: steeply dissected, hilly and rolling terrain |
| | | | | 19–33 (2016) | |
| | | | | 17–32 (2017) | |
| *Paragammana* (**PA**) | *Paragammana*, WL2b, Kegalle | 7.237381, 80.362452 | 2493 | 23–32 (2015) | Reddish brown latasolic soils: steeply dissected, hilly and rolling terrain |
| | | | | 23–32 (2016) | |
| | | | | 22–32 (2017) | |
| *Mawanella* (**MA**) | *Mawanella*, WM2b, Kegalle | 7.250664, 80.452431 | 2469 | 18–31 (2015) | Red yellow podzolic soils: steeply dissected, hilly and rolling terrain |
| | | | | 19–33 (2016) | |
| | | | | 17–32 (2017) | |
| *Rambukkana* (**RA**) | *Rambukkana*, WL2b, Kegalle | 7.323879, 80.398160 | 2281 | 19–31 (2015) | Red yellow podzolic soils: steeply dissected, hilly and rolling terrain |
| | | | | 20–35 (2016) | |
| | | | | 18–34 (2017) | |
| *Polonnaruwa-Supun* (**PS**) | *Nirdeshagama*, DL1c, Polonnaruwa | 7.880910, 80.940870 | 1678 | 20–35 (2015) | Reddish brown earths and low humic gley soils: undulating terrain |
| | | | | 21–36 (2016) | |
| | | | | 19–36 (2017) | |

The mean annual rainfall and temperature values are given based on the records available for the nearest meteorological station of the Department of Agriculture, Sri Lanka

## Fruit size measurements

The fresh fruit weight (g), length (cm), width (cm) perimeter (cm), and diameter (cm) were measured from 12 fruits per accession in each year. The shell thickness (cm) was obtained from each fruit at six different random places using a Vernier caliper. The pulp, shell, and seed related parameters were measured from five fruits per accession in each year. The pulp (including seeds) was scraped out and collected using a tablespoon from each fruit separately and weighed in grams. The shell weight (g) was also measured. The percentage weight of pulp with seeds, percentage weight of shell, and pulp to shell ratio were calculated. The pulp was excavated using a spatula to pick seeds. The number of seeds per fruit was counted. The weight of seeds (g) per fruit was also measured. The net pulp weight (i.e., pulp weight without seeds), percentage weight of net pulp, and mean-number of fresh fruits at the golden-ripe stage from each accession that is required to harvest 1 kg of net pulp were also calculated.

## Fruit size data analysis

The size measurements, shell thickness, and the number of seeds were subjected to the General Linear Model (GLM) Procedure in Statistical Package SAS 9.4 (SAS Institute, Cary, NC, USA, 2013). The significant mean differences were obtained using the Tukey option. The fruit weight (g), length (cm), perimeter (cm) and inner diameter (cm) were normalized to the range of 0–1 and subjected to Principal Component (PC) analysis using the statistical package Minitab 16 (Minitab Inc., USA). The first two PCs were used to draw the PC biplot to depict the variation of fruit size in bael accessions. The fruit pulp, shell, and seed measurements were subjected to the GLM Procedure and Tukey option in SAS.

## Organoleptic assessment of ripe bael fruits

We conducted the organoleptic panel study on ripe bael fruits using 30 panelists. Each panelist provided written informed consent before participation in the taste panel. The panel ranked

the fruits in a scale of five-levels (1: least preference; 2: low preference; 3: average preference; 4: high preference; 5: highest preference) for the parameters; external appearance, flesh color, aroma, texture, sweetness, and overall preference. The data generated from the taste panel were subjected to FREQ procedure in SAS, and relative preferences were calculated as row percentages for bael accessions. The weighted scores were assigned to each accession for each organoleptic parameter by summing up the products of row percentages and the associated ranks. The weighted scores were subjected to PC analysis in Minitab. The PC biplot was drawn between two major PCs to infer the diversity of accessions based on the organoleptic properties.

## Inductively coupled plasma mass spectrometry (ICP-MS) procedure for elemental analysis

Thirty-one elements, including 13 macro and micronutrients in ripe bael pulp samples, were quantified using the ICP-MS method. The pulp extracted from the fruits collected in 2017 was assessed in triplicate for each accession. The fruit pulp was oven-dried in heat resistant crucibles at 80˚C for three-five days until the constant dry weight was reached. This step made samples into hard rock or dried gum like substance. The samples were crushed using mortar and pestle. After crushing, the semi powdered samples were sieved through a fine mesh to obtain finely powdered samples. Approximately 0.2 grams of powdered samples were subjected to digestion using 5 ml of conc. $HNO_3$ and 2 ml of $H_2O_2$ for 30 minutes at 180˚C using Mars-6 Microwave Digester (CEM; Mathews, NC). As a control, 2 ml of deionized water was subjected to the same procedure. The resulted liquid samples after digestion were filtered through 125 μm filter papers. The colorless liquid filtrate samples obtained were diluted up to 50 ml in volumetric flasks. Five milliliters of aliquots of digested samples and the standard comparison sample were then subjected to ICP-MS using Thermo ICapQ analyzer (Thermo-Fisher Scientific Inc., Bremen, Germany). The elemental contents were calculated using the formula given in van de Wiel, (2003) [18]. The elemental data were subjected to GLM procedure in SAS, and the significant mean differences were obtained using the Tukey option. The elemental data were normalized to the range of 0–1 subjected to PC analysis in Minitab. The two first two PCs and three PCs were separately used to draw the PC biplot and PC triplot, respectively.

## DNA extraction, PCR and sequencing

The genomic DNA was isolated from immature bael leaves using the Qiagen DNeasy Plant Mini Kit (Cat. No.: 69104). The PCR was carried out in the thermal cycler TP600 (Takara Bio., Otsu Shiga, Japan) for four plant DNA barcoding markers (*trnH-psbA*, *atpB-rbcL* spacer, *matk-trnT* spacer, and *trnL*) (S1 Table). The forward and reverse primer details and the PCR conditions are also given in S1 Table. The PCR cocktail (15 μl) contained 5× Go Taq Green Master Mix (7.5 μl), 10 μM forward and reverse primers (0.5 μl each), 10 μM spermidine (3.5 μl) [19], and 0.5 μl of template DNA (60 ng/μl). The DNA from the apple variety, Spartan; and distilled water in place of template were used as the positive and negative controls, respectively. The PCR products were size separated using 2% agarose gel electrophoresis and observed for the presence of expected bands (S1 Fig). The PCR products were purified using the PCR Purification Kit, QIAquick (Cat. No.: 28104). The purified PCR products were subjected to 3× cycle sequencing in the Genetic Analyzer 3500 (Cat. No.: 622–0010, Applied Biosystems).

## Phylogenetic analysis

The raw outputs from the sequencing were initially visualized in MEGA 7 [20] to determine the initial and end noise. Separate alignments were created employing the Clustal W algorithm

[21] for different markers (*matK-trnT*, *atpB-rbcL*, *trnL-trnF* and *trnH-psbA* intergenic) in MEGA 7 [20]. Then the initial and end noise regions were trimmed, and alignment was checked manually as automated methods could add unwanted INDELs into the alignment. Then the four datasets were combined, and the single alignment was exported to PAUP 4.0a [22]. The UPGMA tree was built employing uncorrected pairwise distances, and the gaps were treated by distributing proportionally to non-ambiguous SNPs so that INDELs can be treated equally. All the substitutions were treated equally. The final dendrogram was further modified using FigTree v1.4.3 [23].

The sequences were adapted from Bayer *et al.*, (2009) [16] for *matK-trnT*, *atpB-rbcL* and *trnL-trnF* intergenic spacers (S3 Table) to determine the phylogenetic position of Sri Lankan *A. maemelos* of the Rutaceae phylogeny. Initially, the alignments were created separately for each marker, and finally, the combined dataset was created in MEGA 7 [20]. A tree search was carried out in Maximum Likelihood (ML) formwork in RAxML [24] by carrying out a rapid bootstrap algorithm [25]. The algorithm was run for 1000 iterations, and the evolutionary model to compensate the dataset was chosen as GTRGAMMA [26]. Then the best ML tree was used to imprint the bootstrap (bs) values by drawing all the bipartitions into a single tree topology. The final majority rule consensus tree was further modified using FigTree v1.4.3 [23]. The tree was constructed in the Bayesian framework to support the ML tree. Initially, a model selection was carried out in Akaike information criteria (AIC) [27] and corrected Akaike information criteria (AICc) [28] in the J-model test [29] to infer the best nucleotide substitution model to explain our dataset. Then, the best model parameters were used to carry out the tree construction in the Bayesian framework. Using MrBays [30], four Markov chain Monte Carlo (MCMC) were implemented for 50 million iterations to sample the best trees in the tree space. The analysis was set to retain a tree in every 5000 chains, and the initial 10% of the trees were discarded as burn-in. From all the preserved trees, the 50% majority-rule consensus tree was drawn, and posterior probabilities (PP) from each branch were inferred. Finally, the congruence of Bayesian and ML trees was patterned, and the PP values were added to the nodes of the ML tree. The tree constructions and model selection were carried out in the CIPRES Science Gateway [31].

## Time calibration

The software package BEAST 2.0 [32] was used for the calibration of the divergence times. Since the best nucleotide substitution model was inferred that fits the sequence datasets of the current study, the rate and shape parameters were implemented to carry out the tree search. A relaxed lognormal clock [33] was used as it can be used to define the substitution rates for each branch. The clock rates were kept being estimated since the divergence dating was done by calibrating the internal nodes of the phylogeny. A mechanistic species level process was used as the branching model because the models are more applicable in divergence dating of sample trees with multiple taxa. Thus, the fossilized birth-death (FBD) model was used as it encounters with lineage diversification utilizing both extinction and speciation [34]. Two lognormal priors were used to calculate birth and death rates. Since the present dataset mostly associated subfamily Aurantioideae, the older nodes (Family: Rutaceae) were not calibrated due to the lack of samples on the species tree.

It is essential to calibrate the nodes that include the whole spectrum of species that can be aggregated into the particular clade. Since there is a considerable amount of data in the citrus clade as well as many studies that have been carried out inferring the divergence time of citrus plants, the species tree of *A. marmelos* was calibrated mostly using different clades of the citrus crown. Bayer et al., (2009) [16] divided the citrus clade into two groups, i.e., mostly southern

clade that includes many plants from the Australian biogeographic zone and Northern clade. Many have carried out fossil calibrations of the citrus plants using different approaches [35–37]. Thus, in the present study, we used consensus calibration points according to those studies. The Most Recent Common Ancestor (MRCA) priors were used to calibrate the species tree of *A. marmelos*. The citrus crown was calibrated using fossil calibration MRCA before an upper boundary of 10.3 million years ago (MYA) and a lower boundary of 4.3 MYA. The Northern and Southern clades were calibrated using fossil calibration MRCA priors with a lower boundary of 3.2 MYA and an upper boundary of 11.6 MYA, and 3.5 MYA lower boundary and 8.4 MYA upper boundary, respectively. In time calibrations of species trees, it is essential to calibrate one basal lineage that connects most of the taxa. Thus, the subfamily Aurantioideae crown was calibrated using MRCA fossil calibration prior to 12.1 MYA lower bound and 28.2 MYA upper boundary. Four MCMC trees were implemented to run a heuristic tree search for 50 million iterations with an initial 10% burn-in. TreeAnotator [32] was used to draw the final Maximum Clade Credibility (MCC) tree. The reliability of the tree search was checked by assessing the Effective sample sizes (ESS) and Autocorrelation time using TRACER v1. 4 [38]. The final MCC tree was further modified using FigTree v1.4.3 [23].

## Results

### Variation of the fruit size parameters

The weight, length, width, perimeter, inner diameter, and shell thickness of the fruits were not significantly affected by year (P>0.05). The mean fruit weight was significantly highest in PS (followed by RA and MA) in all three years (P<0.05) (Table 2). Although the mean fruit weight of PS was significantly higher than that of RA, the mean fruit length, width, perimeter, and inner diameter were not significantly different among them (P>0.05). The mean fruit size parameters were markedly lower in BB and PA (Table 2). The mean shell thickness of BB was the highest (0.4 cm) compared to that of all other four accessions (0.3 cm) (P<0.05) (Table 2). Furthermore, all the fruit size parameters followed the same trend throughout the collection periods (2015–2018). The highest net pulp percentage was observed in PS, followed by RA,

**Table 2. Mean fruit size parameters of the bael accessions.**

| Year | Bael accession | Weight (g) | Length (cm) | Width (cm) | Perimeter (cm) | Diameter (cm) | Shell thickness (cm) |
|---|---|---|---|---|---|---|---|
| 2015 | BB | 134.0$^f$±7.16 | 6.1$^{de}$±0.12 | 6.0$^e$±0.12 | 18.9$^{def}$±0.37 | 5.2$^{de}$±0.12 | 0.4$^a$±0.0 |
| | PA | 129.7$^f$±6.67 | 5.9$^e$±0.12 | 5.8d$^e$±0.11 | 18.4$^{ef}$±0.34 | 5.2$^{de}$±0.11 | 0.3$^a$±0.0 |
| | MA | 269.5$^{def}$±11.74 | 7.0$^{cde}$±0.08 | 7.4b$^c$±0.15 | 23.4$^{bcd}$±0.47 | 6.8$^{bc}$±0.15 | 0.3$^a$±0.0 |
| | RA | 605.7$^c$±23.19 | 9.1$^{ab}$±0.17 | 9.3$^a$±0.16 | 29.3$^a$±0.51 | 8.7$^a$±0.16 | 0.3$^a$±0.0 |
| | PS | 992.6$^a$±25.13 | 9.9$^a$±0.22 | 9.4$^a$±0.19 | 29.5$^a$±0.60 | 8.8$^a$±0.19 | 0.3$^a$±0.0 |
| 2016 | BB | 157.3$^{def}$±7.61 | 5.9$^e$±0.17 | 6.0$^e$±0.17 | 18.9$^{cdef}$±0.55 | 5.2$^{de}$±0.17 | 0.4$^a$±0.0 |
| | PA | 149.7$^{def}$±13.66 | 6.1$^{de}$±0.17 | 6.0$^e$±0.18 | 18.9$^{ef}$±0.58 | 5.4$^{de}$±0.18 | 0.3$^a$±0.0 |
| | MA | 309.1$^{de}$±8.92 | 7.4$^{cd}$±0.14 | 7.5b$^c$±0.19 | 23.4$^{bc}$±0.61 | 6.9$^{bc}$±0.19 | 0.3$^a$±0.0 |
| | RA | 858.3$^{ab}$±25.83 | 9.0$^{ab}$±0.20 | 8.6$^{ab}$±0.18 | 26.9$^{ab}$±0.55 | 8.0$^{ab}$±0.18 | 0.3$^a$±0.0 |
| | PS | 916.1$^{ab}$±45.39 | 10.0$^a$±0.24 | 9.5$^a$±0.26 | 29.9$^a$±0.81 | 8.9$^a$±0.26 | 0.3$^a$±0.0 |
| 2017 | BB | 137.2$^{ef}$±5.58 | 6.1$^{de}$±0.13 | 6.0$^{cde}$±0.13 | 18.9$^{cdef}$±0.41 | 5.2$^{de}$±0.13 | 0.4$^a$±0.0 |
| | PA | 170.4$^{def}$±13.68 | 5.6$^e$±0.17 | 5.3$^e$±0.19 | 16.6$^f$±0.60 | 4.7$^e$±0.19 | 0.3$^a$±0.0 |
| | MA | 313.9$^d$±8.66 | 7.8$^{bc}$±0.11 | 7.1$^{cd}$±0.16 | 22.3$^{cde}$±0.51 | 6.5$^{cd}$±0.16 | 0.3$^a$±0.0 |
| | RA | 757.2$^{bc}$±32.08 | 9.9$^a$±0.40 | 8.6$^{ab}$±0.22 | 26.9$^{ab}$±0.68 | 8.0$^a$±0.22 | 0.3$^a$±0.0 |
| | PS | 992.2$^a$±20.69 | 10.3$^a$±0.23 | 9.6$^a$±0.19 | 30.1$^a$±0.60 | 9.0$^a$±0.19 | 0.3$^a$±0.0 |

Means denoted by same letters within each column are not significantly different at P<0.05.

**Table 3. Mean weights of pulp, shell and seeds of the fruits.**

| Year | Accession | Pulp with seeds–Weight (g) | Shell weight (g) | Pulp with seeds (%) weight basis | Shell percentage | Pulp to shell ratio | No. of seeds | Seed weight/ fruit (g) | Net pulp weight/fruit (g) | Net pulp (%) | No. of fruits needed for 1kg of pulp[#] |
|---|---|---|---|---|---|---|---|---|---|---|---|
| 2015 | BB | 73.50[de] ±3.75 | 68.11[g] ±3.47 | 52.76[d] ±2.69 | 47.68[ab] ±2.43 | 1.01[ef] ±0.05 | 45.4[cd] ±2.31 | 43.85[cdef] ±2.24 | 26.54[d] ±1.35 | 19.97[f] ±2.28 | 37.7 |
|  | MA | 199.97[c] ±10.20 | 118.33[cde] ±6.03 | 63.94[bcd] ±3.26 | 38.61[bc] ±1.88 | 1.59[def] ±0.08 | 57.6[bcd] ±2.93 | 58.68[ab] ±2.99 | 129.09[cd] ±6.58 | 43.24[cd] ±2.21 | 7.7 |
|  | PA | 78.01[cd]e ±3.98 | 83.86[efg] ±4.28 | 49.03[d] ±2.50 | 51.35[a] ±2.62 | 0.87[f] ±0.04 | 43.4[d] ±2.21 | 33.11[fg] ±1.69 | 40.78[cd] ±2.08 | 26.84[ef] ±1.37 | 24.5 |
|  | RA | 532.66[b] ±27.16 | 147.67[bc] ±7.53 | 79.09[abc] ±4.03 | 21.37[d] ±1.09 | 3.36[c] ±0.17 | 56.4[abcd] ±2.88 | 47.21[bcd] ±2.41 | 446.26[b] ±22.75 | 69.41[b] ±3.54 | 2.2 |
|  | PS | 796.42[a] ±40.61 | 186.30[a] ±9.50 | 83.12a ±4.24 | 18.95d ±0.97 | 4.02[abc] ±0.21 | 18.0[e] ±0.92 | 14.30[h] ±0.73 | 720.02[a] ±36.71 | 78.71[ab] ±4.01 | 1.4 |
| 2016 | BB | 71.50[e] ±3.65 | 67.50[g] ±3.44 | 51.97[d] ±2.58 | 49.92[a] ±2.55 | 1.14[def] ±0.06 | 48.8[bcd] ±2.48 | 45.42[cde] ±2.32 | 29.68[cd] ±1.51 | 22.13[f] ±1.13 | 33.7 |
|  | MA | 194.53[cd] ±9.92 | 117.27[cde] ±5.98 | 62.98[cd] ±3.12 | 38.61[bc] ±1.97 | 1.79[d] ±0.09 | 62.0[ab] ±3.15 | 60.78[a] ±3.10 | 144.36[c] ±7.36 | 29.75[def] ±2.44 | 6.9 |
|  | PA | 75.89[de] ±3.87 | 83.11[efg] ±4.24 | 48.30[d] ±2.40 | 53.76[a] ±2.74 | 0.98[ef] ±0.05 | 46.8[cd] ±2.38 | 34.29[efg] ±1.75 | 45.60[cd] ±2.33 | 29.75[def] ±1.52 | 21.9 |
|  | RA | 518.17[b] ±26.42 | 146.34[bcd] ±7.46 | 77.90[abc] ±3.86 | 22.38[d] ±1.14 | 3.79[abc] ±0.19 | 61.0[ab] ±3.10 | 48.90[abcd] ±2.49 | 499.06[b] ±25.45 | 76.92[ab] ±3.92 | 2.0 |
|  | PS | 774.75a ±39.50 | 184.64[a] ±9.41 | 81.87[ab] ±4.06 | 19.84[d] ±1.01 | 4.54[a] ±0.23 | 19.6[a] ±0.99 | 14.81[h] ±0.76 | 805.21[a] ±41.06 | 87.22[a] ±4.45 | 1.2 |
| 2017 | BB | 69.93[e] ±3.57 | 64.35[g] ±3.28 | 50.55[d] ±2.65 | 46.98[abc] ±2.40 | 1.05[def] ±0.05 | 50.8[abcd] ±2.58 | 41.38[defg] ±2.11 | 28.04[d] ±1.43 | 18.84[f] ±0.96 | 35.7 |
|  | MA | 190.26[cde] ±9.70 | 111.79[def] ±5.70 | 61.26[cd] ±3.21 | 36.33[c] ±1.85 | 1.65[de] ±0.08 | 64.4[a] ±3.28 | 55.38[abc] ±2.82 | 136.39[cd] ±6.95 | 40.80[cde] ±2.08 | 7.3 |
|  | PA | 74.22[de] ±3.78 | 79.23[fg] ±4.04 | 46.98[d] ±2.46 | 50.60[a] ±2.58 | 0.87[f] ±0.05 | 48.4[bcd] ±2.47 | 31.24[g] ±1.59 | 43.08[cd] ±2.20 | 25.32[def] ±1.29 | 23.2 |
|  | RA | 506.79[b] ±25.84 | 139.51[cd] ±7.11 | 75.78[abc] ±3.97 | 21.06[d] ±1.07 | 3.49[bc] ±0.18 | 63.2[a] ±3.22 | 44.55[cdef] ±2.27 | 471.48[b] ±24.04 | 65.48[b] ±3.34 | 2.1 |
|  | PS | 757.74[a] ±38.64 | 176.02[ab] ±8.98 | 79.63[abc] ±4.17 | 18.67[d] ±0.95 | 4.18[ab] ±0.21 | 20.0[e] ±1.03 | 13.49[h] ±0.69 | 760.71[a] ±38.79 | 74.26[ab] ±3.79 | 1.3 |

Means denoted by same letters within each column are not significantly different at P<0.05.

indicating the superiority of these two accessions compared to the other three (Table 3). Fig 1 illustrates the images of the representative fruits of five bael accessions alongside with the longitudinal and cross-sections. The mean number of seeds was significantly lowest in PS and highest in MA and RA (P<0.05) (Table 3). If the data of three years pooled together, the estimated numbers of fruits needed to harvest 1 kg of net-pulp were 35.7 fruits for BB, 7.3 for MA, 23.2 for PA, 1.3 for PS, and 2.1 for RA. The seeds observed in PS are shrunken, smaller and sterile, and all the other accessions possessed fully developed and fertile seeds. The data of fruit size, along with the rest of the data sets collected in the present study are given in S2 Table.

The PCA for the fruit size parameters, weight, length, width, and diameter yielded three principal components (PC) to explain the 100% variance (S4 Table). The PC loading status in terms of eigenvalues is shown as a scree plot given in S2A Fig. The weight and length were separately associated with the overall variance of fruit size. In contrast, circumference, width, and diameter got the same effect on fruit size variance because of the near-spherical shape of the bael fruits (S2B Fig). In the PC biplot drawn among PC1 and PC2, two distinct clusters could be seen for the fruit size. One cluster contains PS and RA fruits; however, the relatively larger size of PS fruits is apparent due to the clear separation. The other cluster holds MA, PA, and

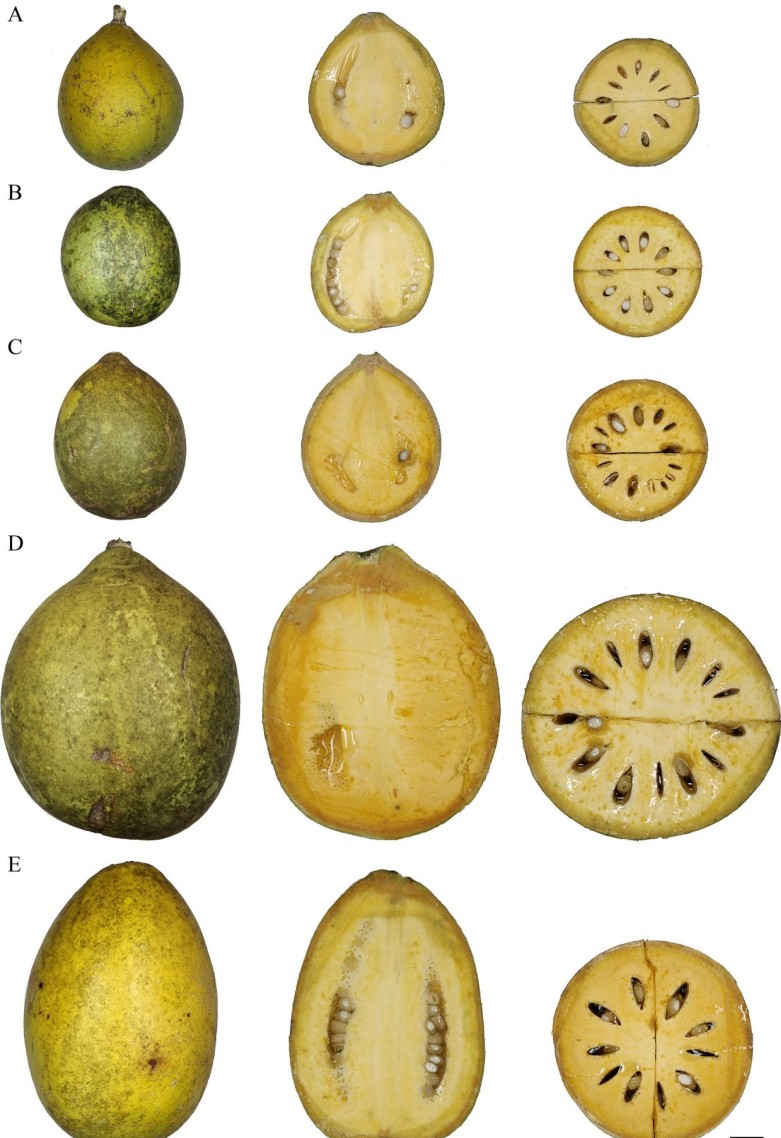

**Fig 1. The appearance of the representative fruits of the elite bael accessions.** A: BB; B: PA; C:MA; D: RA; E: PS. The side-view, longitudinal and cross sections are shown from the same fruit for each accession. Scale bar: 1 cm.

BB, which have relatively smaller fruits. However, it is evident from the separation that MA fruits are bigger than the BB and PA fruits (Fig 2).

## Organoleptic preference on bael fruits/accessions

The organoleptic parameters; external appearance, flesh color, aroma, texture (an indicator of flesh to fiber ratio), sweetness and overall preference (visual appeal and the taste) were significantly associated with the type of accession ($\chi$2 is ranging from 60.0–368.7; P<0.05). The highest preferred external appearance was observed for PS, followed by RA. The PA and BB fruits were not highly preferred (Fig 3A). The most sorted flesh color was reported for RA, and all accessions got ranked above average for the flesh color (Fig 3B). The most appealing aroma

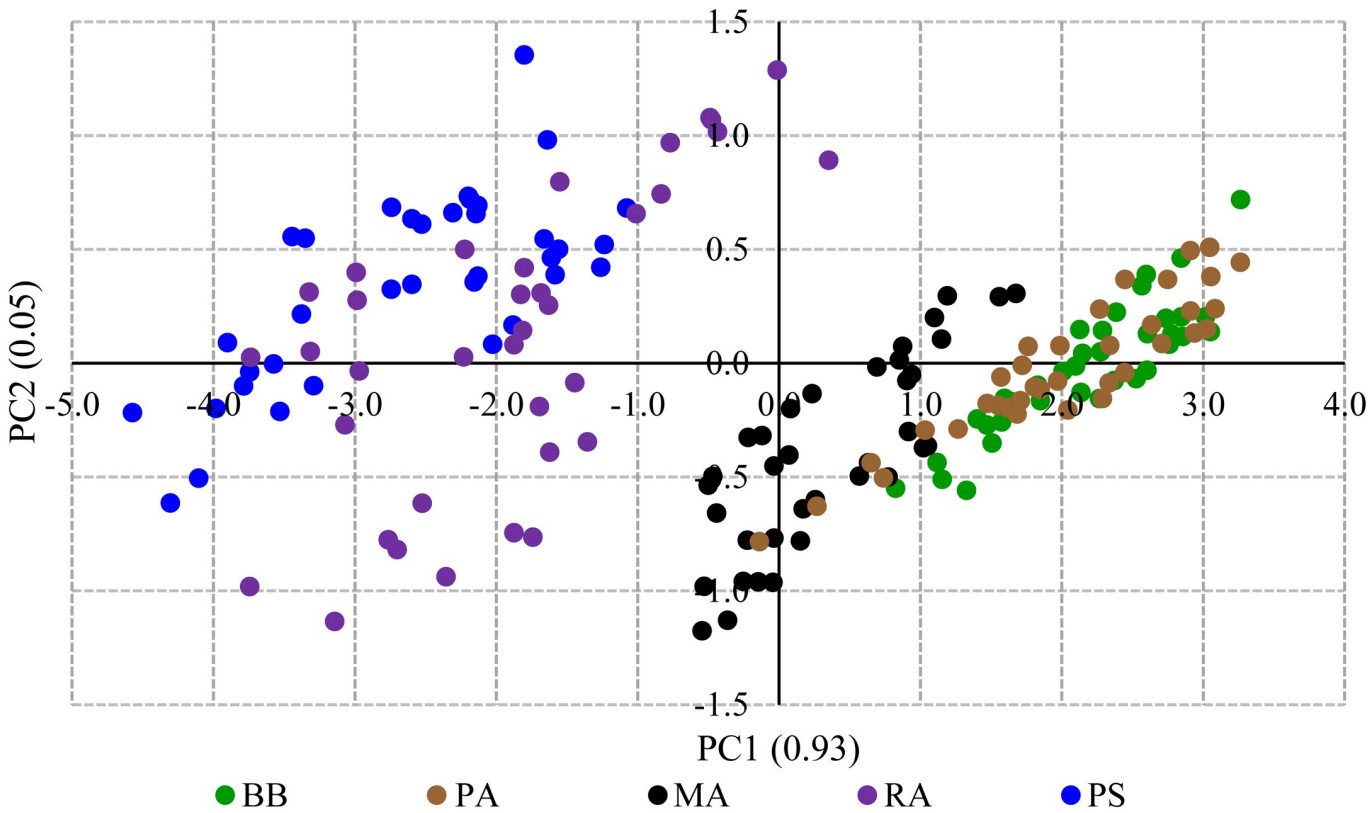

**Fig 2. The PC biplot drawn based on the two major PCs calculated for fruit size parameters.** The contribution of the PC to the total variance is given in parenthesis.

was reported for RA, and like flesh color, all bael accessions were ranked above average for aroma (Fig 3C). The texture of BB was the least preferred as 40%, and 60% of the panelists ranked BB texture as the least and low, respectively (Fig 3D). The MA and PA fruits were reported for above-average sweetness, whereas all tasters ranked PS for the average sweetness. The best-preferred sweetness was reported for RA by all tasters (Fig 3E). When considering the overall preference, RA was the highest preferred accession, followed by PS. The accessions MA and PA received the average preference, whereas BB received below average overall preference (Fig 3F).

The ratings received by each accession for all the organoleptic parameters were converted to weighted scores and subjected to PCA which defined four PCs to explain the entire variability of the organoleptic parameters. The PC loading status and Eigenvalues for the weighted scores calculated based on organoleptic parameters are given in S6 Table and S3A Fig. The first two PCs explained 90% of the variability. The PC biplot placed accessions distantly (Fig 4). The PC biplot for the weighted scores of organoleptic parameters provided a clear separation of the bael accessions to depict their strengths for consumer preference. All the organoleptic parameters exhibited their individual effects on the overall variance of organoleptic properties. However, the aroma and sweetness got positioned adjacently in the loading plot indicating their interrelatedness (S3B Fig). The preferred flesh color got placed distantly from the other organoleptic parameters suggesting that it has a peculiar effect on the preference for bael fruits.

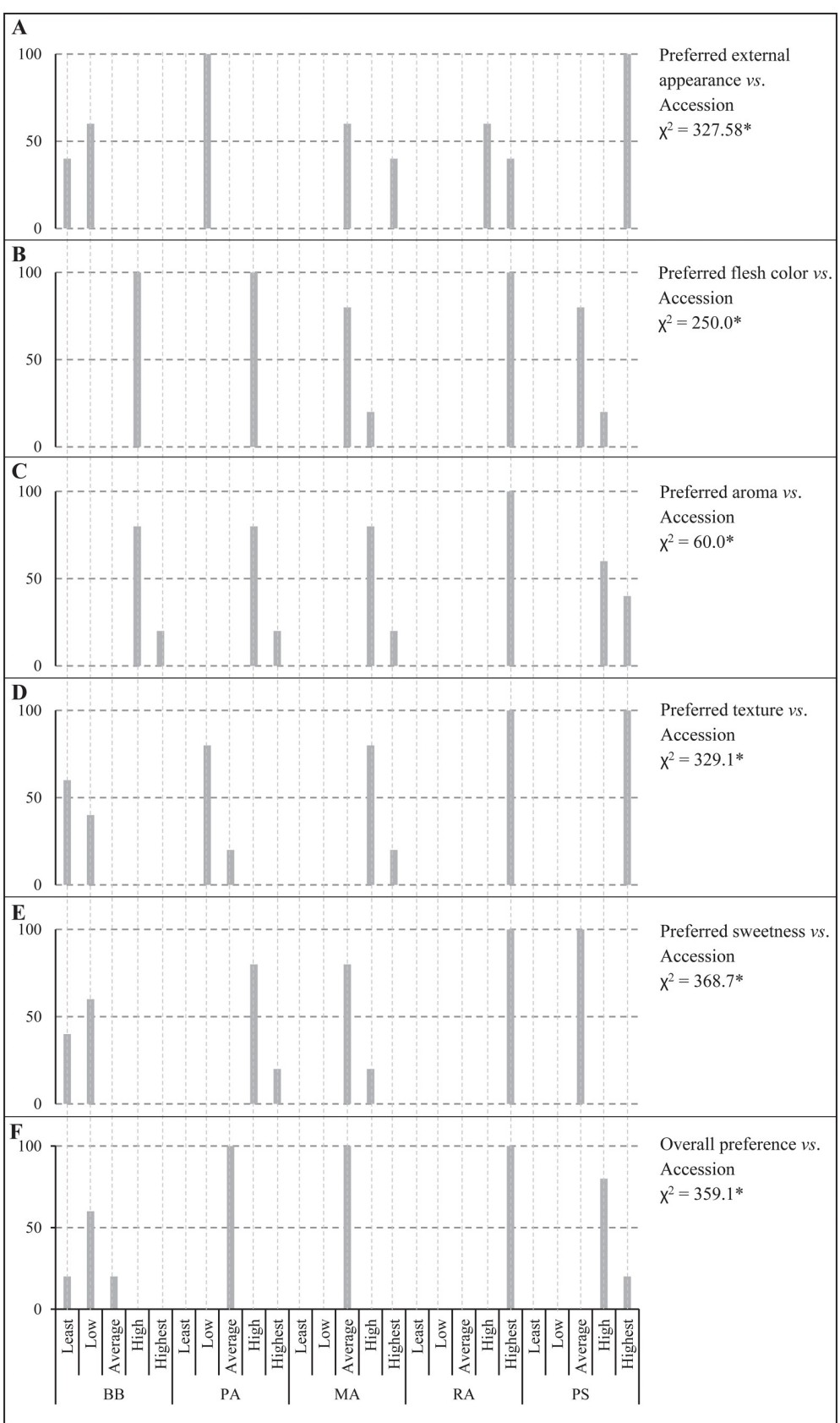

**Fig 3. Association analysis for organoleptic parameters and the ripe fruits of the bael accessions.** The same scales and values of X and Y axes are shown. The X-axis is shown only once. The vertical grid lines are included to clearly display the preference level classes on the X-axis. Y-axis indicates the percentage respondents.

## Variation of the elemental contents in bael fruit pulp

The elemental profiles revealed by ICP-MS for five bael accessions are given in Table 4. The ICP-MS analysis has not detected Na in bael. The significantly highest Mg content was detected in ripe bael fruits of PA (8881.38 mg/kg), and the least amount was detected in RA (485.83 mg/kg). The significantly highest content of K was detected in MA (15607.70 mg/kg). When the quantitative data of 36 elements were subjected to PCA, a total of 13 PCs resulted. The first and second PCs explained 53%, and the first five PC explained 85% of the total variance (S6 Table; S4A Fig). The PC loading plot for PCA of the elemental contents shows the unique efforts and potential close associations in the elemental contents of the ripe fruit pulp (S4B and S4C Fig). Except for Se isotopes, the effects of all the isotope bearing elements assessed (Pb and Fe), positioned in the same area indicating that their effects on overall diversity are similar or the contents of isotopes of these elements are interrelated. We examined PC biplot and PC triplot to infer the relationships among the bael accessions based on the ICP-MS elemental contents of ripe fruits (Fig 5 and S4C Fig). The samples assessed for each accession showed a variation; however, three clusters were apparent (cluster 1: PS; cluster 2: RA; cluster 3: MA, PA, and BB) (S4C Fig).

## Phylogenetic analysis

The UPGMA tree drawn for combined datasets was separated into two clusters at 0.48% separating BB and MA with PA, PS, and RA (Fig 6). BB and MA had 0.094% of uncorrected pairwise genetic divergence, whereas PA diverged out from PS and RA at 0.042% of uncorrected pairwise genetic divergence. PS and RA claded sister to each other, having shallow divergence (with 0.79% uncorrected pairwise genetic divergence). The model selection resulted the GTR +I+G model (AC = 0.850, AG = 1.266, AT = 0.348, CG = 0.688, CT = 0,995, GT = 1.000; proportion of invariable sites = 0.34; Gamma shape parameter = 1.04). The majority-rule

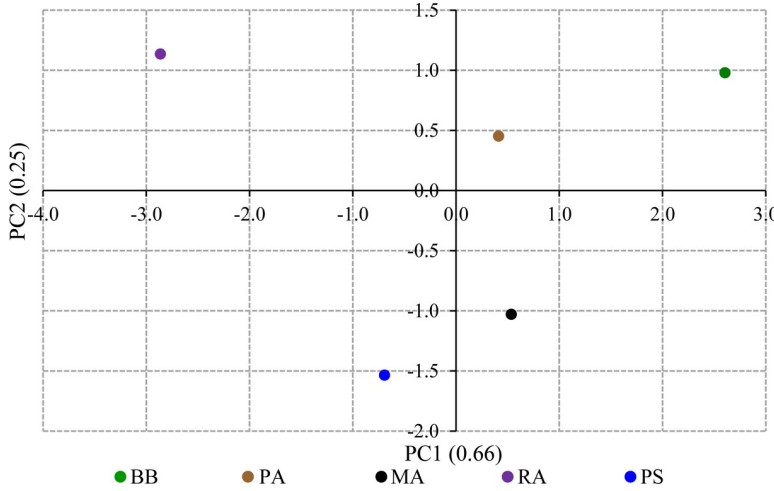

**Fig 4. The PC biplot for the weighted scores of organoleptic parameters.** The contribution of the PC to the total variance is given in parenthesis.

**Table 4. Mean elemental contents in ripe fruit samples of elite bael accessions measured with ICP-MS.**

| Element | BB | PA | MA | RA | PS |
|---|---|---|---|---|---|
| Li$^7$ (ppb) | $^{bc}7.30\times10^{-3}\pm6.55\times10^{-4}$ | $^{ab}8.93\times10^{-3}\pm3.06\times10^{-4}$ | $^{cd}3.17\times10^{-3}\pm1.39\times10^{-3}$ | $^{d}6.35\times10^{-4}\pm5.18\times10^{-4}$ | $^{a}1.29\times10^{-2}\pm1.05\times10^{-3}$ |
| Be$^9$ (ppb) | $^{a}5.95\times10^{-4}\pm3.54\times10^{-5}$ | $^{ab}3.44\times10^{-4}\pm8.35\times10^{-5}$ | $^{b}1.26\times10^{-4}\pm1.03\times10^{-4}$ | $^{b}1.17\times10^{-4}\pm9.56\times10^{-6}$ | $^{a}6.19\times10^{-4}\pm1.04\times10^{-4}$ |
| Na$^{23}$ (ppm) | $^{a}0.00$ | $^{a}0.00$ | $^{a}0.00$ | $^{a}0.00$ | $^{a}0.00$ |
| Mg$^{24}$ (ppm) | $^{b}6.68\times10^{2}\pm1.72\times10^{1}$ | $^{a}8.81\times10^{2}\pm2.56\times10^{1}$ | $^{c}5.05\times10^{2}\pm1.39\times10^{1}$ | $^{c}4.86\times10^{2}\pm1.96\times10^{1}$ | $^{c}4.74\times10^{2}\pm5.81$ |
| Al$^{27}$ (ppb) | $^{a}4.30\pm1.36$ | $^{a}5.61\times10^{-1}\pm4.58\times10^{-1}$ | $^{a}9.30\pm3.85$ | $^{a}4.26\times10^{-1}\pm1.75\times10^{-1}$ | $^{a}8.88\times10^{-1}\pm3.68\times10^{-1}$ |
| K$^{39}$ (ppm) | $^{c}9.73\times10^{3}\pm2.44\times10^{2}$ | $^{c}8.74\times10^{3}\pm2.91\times10^{2}$ | $^{a}1.56\times10^{4}\pm4.79\times10^{2}$ | $^{b}1.28\times10^{4}\pm3.77\times10^{2}$ | $^{b}1.37\times10^{4}\pm7.67\times10^{1}$ |
| Ca$^{44}$ (ppm) | $^{a}2.28\times10^{3}\pm4.52\times10^{1}$ | $^{ab}2.12\times10^{3}\pm1.40\times10^{2}$ | $^{a}2.48\times10^{3}\pm4.16\times10^{2}$ | $^{ab}1.35\times10^{3}\pm1.98\times10^{2}$ | $^{b}8.75\times10^{2}\pm2.01\times10^{2}$ |
| Ti$^{48}$ (ppb) | $^{ab}4.94\pm7.48\times10^{-2}$ | $^{abc}4.57\pm3.04\times10^{-1}$ | $^{a}5.90\pm7.90\times10^{-1}$ | $^{bc}2.95\pm4.64\times10^{-1}$ | $^{c}2.06\pm4.42\times10^{-1}$ |
| V$^{51}$ (ppb) | $^{a}3.56\times10^{-3}\pm7.95\times10^{-4}$ | $^{a}1.47\times10^{-3}\pm1.20\times10^{-3}$ | $^{a}1.70\times10^{-2}\pm5.24\times10^{-3}$ | $^{a}1.73\times10^{-3}\pm1.42\times10^{-3}$ | $^{a}1.18\times10^{-2}\pm5.24\times10^{-3}$ |
| Cr$^{52}$ (ppb) | $^{a}1.63\times10^{-1}\pm7.62\times10^{-2}$ | $^{a}1.14\times10^{-1}\pm9.30\times10^{-2}$ | $^{a}5.54\times10^{-2}\pm4.52\times10^{-2}$ | $^{a}1.85\times10^{-1}\pm1.51\times10^{-1}$ | $^{a}1.49\pm7.39\times10^{-1}$ |
| Mn$^{55}$ (ppb) | $^{b}2.02\pm1.03\times10^{-1}$ | $^{ab}2.11\pm4.03\times10^{-2}$ | $^{ab}2.07\pm1.32\times10^{-1}$ | $^{a}2.62\pm1.28\times10^{-1}$ | $^{b}1.59\pm4.76\times10^{-2}$ |
| Fe$^{56}$ (ppb) | $^{b}3.60\pm1.21$ | $^{b}2.95\pm2.41$ | $^{ab}2.02\times10^{1}\pm1.47$ | $^{a}4.77\times10^{1}\pm6.14$ | $^{ab}2.86\times10^{1}\pm9.68$ |
| Fe$^{57}$ (ppb) | $^{b}4.60\pm1.18$ | $^{b}3.48\pm2.54$ | $^{ab}2.09\times10^{1}\pm1.39$ | $^{a}4.66\times10^{2}\pm6.03$ | $^{ab}2.80\times10^{1}\pm9.15$ |
| Co$^{59}$ (ppb) | $^{b}6.33\times10^{-4}\pm5.17\times10^{-4}$ | $^{b}1.54\times10^{-3}\pm1.26\times10^{-4}$ | $^{b}5.04\times10^{-3}\pm3.30\times10^{-3}$ | $^{a}2.68\times10^{-2}\pm3.15\times10^{-3}$ | $^{a}2.61\times10^{-2}\pm1.92\times10^{-3}$ |
| Ni$^{60}$ (ppb) | $^{bc}1.90\pm4.51\times10^{-1}$ | $^{c}8.16\times10^{-1}\pm1.32\times10^{-1}$ | $^{b}2.61\pm1.52\times10^{-1}$ | $^{b}3.45\pm2.71\times10^{-1}$ | $^{a}5.19\pm3.38\times10^{-1}$ |
| Cu$^{63}$ (ppb) | $^{b}2.81\pm2.36\times10^{-1}$ | $^{b}1.67\pm1.35\times10^{-1}$ | $^{b}5.26\pm2.82\times10^{-1}$ | $^{a}1.54\times10^{1}\pm1.03$ | $^{a}1.51\times10^{1}\pm2.57$ |
| Zn$^{66}$ (ppb) | $^{a}5.30\pm1.39$ | $^{a}2.19\pm1.78$ | $^{a}1.95\times10^{1}\pm7.11$ | $^{a}4.91\pm4.01$ | $^{a}8.40\pm6.86$ |
| Ga$^{71}$ (ppb) | $^{a}6.59\times10^{-3}\pm4.85\times10^{-4}$ | $^{ab}4.81\times10^{-3}\pm5.94\times10^{-4}$ | $^{ab}3.67\times10^{-3}\pm1.27\times10^{-3}$ | $^{b}2.06\times10^{-3}\pm7.91\times10^{-4}$ | $^{ab}4.79\times10^{-3}\pm3.11\times10^{-4}$ |
| As$^{75}$ (ppb) | $^{a}0.00$ | $^{a}0.00$ | $^{a}2.66\times10^{-3}\pm2.17\times10^{-3}$ | $^{a}0.00$ | $^{a}0.00$ |
| Se$^{77}$ (ppb) | $^{a}4.25\times10^{-2}\pm3.11\times10^{-3}$ | $^{ab}3.90\times10^{-2}\pm3.33\times10^{-3}$ | $^{b}2.50\times10^{-2}\pm2.62\times10^{-3}$ | $^{ab}2.42\times10^{-2}\pm1.31\times10^{-3}$ | $^{ab}2.91\times10^{-2}\pm3.38\times10^{-3}$ |
| Se$^{78}$ (ppb) | $^{ab}2.15\times10^{-2}\pm3.30\times10^{-3}$ | $^{ab}1.93\times10^{-2}\pm2.89\times10^{-3}$ | $^{b}1.77\times10^{-2}\pm1.88\times10^{-3}$ | $^{ab}2.04\times10^{-2}\pm3.11\times10^{-3}$ | $^{a}3.70\times10^{-2}\pm4.45\times10^{-3}$ |
| Se$^{82}$ (ppb) | $^{a}3.14\times10^{-2}\pm7.57\times10^{-3}$ | $^{a}1.63\times10^{-2}\pm9.49\times10^{-3}$ | $^{a}6.83\times10^{-2}\pm2.09\times10^{-2}$ | $^{a}1.92\times10^{-2}\pm1.13\times10^{-2}$ | $^{a}4.44\times10^{-2}\pm2.84\times10^{-2}$ |
| Rb$^{85}$ (ppb) | $^{b}2.03\times10^{1}\pm5.12\times10^{-1}$ | $^{c}1.39\times10^{1}\pm5.04\times10^{-1}$ | $^{a}2.37\times10^{1}\pm8.04\times10^{-1}$ | $^{c}1.56\times10^{1}\pm3.59\times10^{-1}$ | $^{d}8.37\times10^{1}\pm1.30\times10^{-1}$ |
| Sr$^{88}$ (ppb) | $^{b}1.21\times10^{1}\pm1.76\times10^{-1}$ | $^{a}1.46\times10^{1}\pm4.61\times10^{-1}$ | $^{ab}1.31\times10^{1}\pm5.39\times10^{-1}$ | $^{c}7.22\pm2.39\times10^{-1}$ | $^{c}7.94\pm6.02\times10^{-2}$ |
| Ag$^{107}$ (ppb) | $^{a}4.01\times10^{-3}\pm3.27\times10^{-3}$ | $^{a}5.19\times10^{-4}\pm3.08\times10^{-4}$ | $^{a}2.55\times10^{-3}\pm1.40\times10^{-3}$ | $^{a}8.15\times10^{-4}\pm6.65\times10^{-4}$ | $^{a}0.00$ |
| Cd$^{111}$ (ppb) | $^{a}1.48\times10^{-3}\pm1.21\times10^{-3}$ | $^{a}0.00$ | $^{a}0.00$ | $^{a}0.00$ | $^{a}1.02\times10^{-3}\pm8.31\times10^{-4}$ |
| In$^{115}$ (ppb) | $^{a}7.32\times10^{-5}\pm4.93\times10^{-5}$ | $^{a}0.00$ | $^{a}3.53\times10^{-5}\pm2.65\times10^{-5}$ | $^{a}0.00$ | $^{a}1.68\times10^{-4}\pm5.83\times10^{-5}$ |
| Cs$^{133}$ (ppb) | $^{a}1.84\times10^{-2}\pm5.48\times10^{-4}$ | $^{c}9.45\times10^{-3}\pm5.13\times10^{-4}$ | $^{b}1.39\times10^{-2}\pm3.29\times10^{-4}$ | $^{b}1.39\times10^{-2}\pm4.58\times10^{-4}$ | $^{d}2.76\times10^{-3}\pm1.67\times10^{-4}$ |
| Ba$^{137}$ (ppb) | $^{b}2.77\times10^{1}\pm1.24$ | $^{a}4.56\times10^{1}\pm6.08\times10^{-1}$ | $^{d}5.77\times10^{-1}\pm1.10\times10^{-1}$ | $^{d}8.53\times10^{-1}\pm1.74\times10^{-1}$ | $^{c}1.36\times10^{1}\pm2.29\times10^{-1}$ |
| Hg$^{202}$ (ppb) | $^{a}1.06\times10^{-3}\pm4.33\times10^{-4}$ | $^{a}6.79\times10^{-4}\pm1.47\times10^{-4}$ | $^{a}8.13\times10^{-4}\pm4.98\times10^{-5}$ | $^{a}7.88\times10^{-4}\pm4.10\times10^{-4}$ | $^{a}5.65\times10^{-4}\pm3.00\times10^{-4}$ |
| Tl$^{205}$ (ppb) | $^{ab}1.72\times10^{-3}\pm7.01\times10^{-4}$ | $^{a}1.92\times10^{-3}\pm5.21\times10^{-5}$ | $^{ab}1.00\times10^{-3}\pm1.66\times10^{-4}$ | $^{ab}5.91\times10^{-4}\pm1.98\times10^{-4}$ | $^{b}0.00$ |
| Pb$^{206}$ (ppb) | $^{a}0.00$ | $^{a}0.00$ | $^{a}5.88\times10^{-2}\pm4.46\times10^{-2}$ | $^{a}0.00$ | $^{a}0.00$ |
| Pb$^{207}$ (ppb) | $^{a}0.00$ | $^{a}0.00$ | $^{a}5.31\times10^{-2}\pm4.29\times10^{-2}$ | $^{a}0.00$ | $^{a}0.00$ |
| Pb$^{208}$ (ppb) | $^{a}0.00$ | $^{a}0.00$ | $^{a}5.61\times10^{-2}\pm4.25\times10^{-2}$ | $^{a}6.1\times10^{-3}\pm5.04\times10^{-3}$ | $^{a}0.00$ |
| Bi$^{209}$ (ppb) | $^{a}4.58\times10^{-4}\pm3.97\times10^{-5}$ | $^{a}9.31\times10^{-4}\pm6.86\times10^{-4}$ | $^{a}1.17\times10^{-3}\pm1.99\times10^{-4}$ | $^{a}2.09\times10^{-4}\pm1.71\times10^{-4}$ | $^{a}9.78\times10^{-4}\pm3.72\times10^{-4}$ |
| U$^{238}$ (ppb) | $^{a}4.19\times10^{-4}\pm3.42\times10^{-4}$ | $^{a}0.00$ | $^{a}1.19\times10^{-3}\pm5.27\times10^{-4}$ | $^{a}0.00$ | $^{a}1.28\times10^{-3}\pm6.78\times10^{-4}$ |

Means denoted by same letters within rows are not significantly different at $P<0.05$

consensus tree built in the ML framework had an almost similar topology with a 50% majority rule consensus tree constructed in the Bayesian framework. The higher bs and PP support values reinforced most of the branches. The nodes that had lower bs were supported with higher PP or vice versa (Fig 7). A highly supported clade was formed with *A. marmelos* sequences of the present study and the Indian *A. marmelos* sequence. However, the Sri Lankan bael accessions clustered sister to the Indian *A. marmelos*. The clade containing the members of *A. marmelos* clustered sister to *Afraegle paniculata*, which is found in Africa (bs = 100; PP = 95).

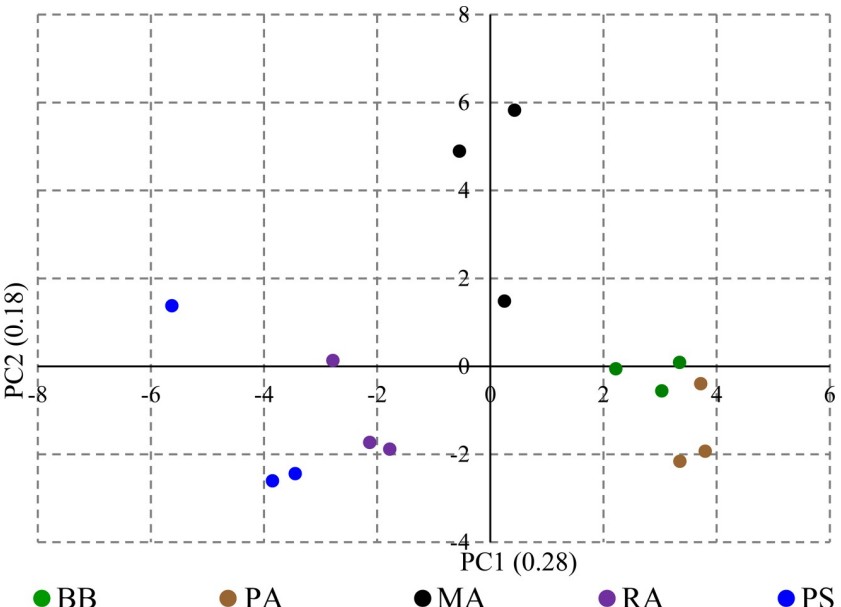

**Fig 5. The PC biplot based on ICP-MS elemental data.** The contribution of the PC to the total variance is given in parenthesis.

### Divergence dating

The ESS for all the priors was above 200, indicating that the probing of the tree-space was done independently by the MCMC chains. In the tree search, the initial 500,000 trees were discarded as burn-in. The final tree had the mean log-likelihood value of -10779.38. The clock rate was estimated as $8.82 \times 10^{-4}$. The time-calibrated tree had an almost similar branching pattern with the ML and Bayesian trees constructed. Our time analysis suggested that separation of *Afraegle paniculata* from *A. marmelos* happened in late Miocene epoch (9.441 MYA: 6.986– 11.776, 95% HDP). The *Aegle/Afraegle* clade separated from *Aeglopsis chevalieri* 10.004 MYA [6.522–12.678, 95% HDP]. The separation between Indian bael and Sri Lankan bael was estimated to be 8.5165 MYA (2.624–5.677 95% HDP) (Fig 8).

### Discussion

The FCRDS of Sri Lanka has selected five elite accessions of bael for distribution as cultivars within the country. The accessions BB, PA, MA, and RA were chosen from the Wet Zone, and

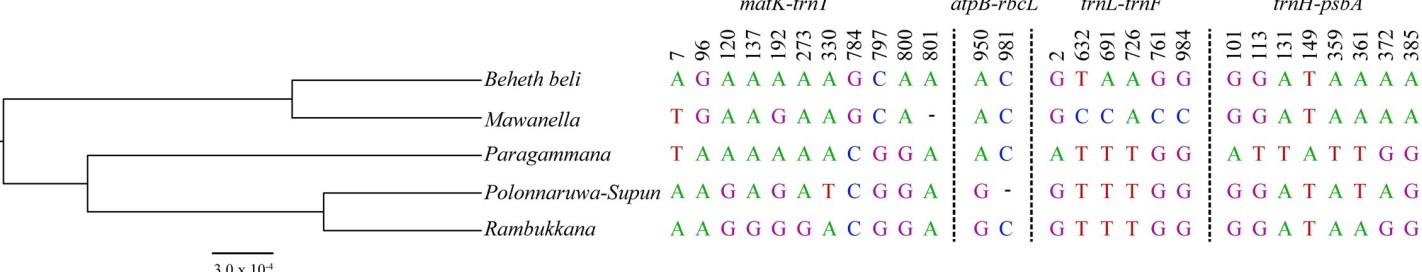

**Fig 6. The hierarchical clustering of the accessions of *A. marmelos* using combined sequence datasets.** The unweighted pair group method with arithmetic mean (UPGMA) tree is given in the left. The scale bar represents the uncorrected pairwise distance. The associated SNP and INDEL positions for *matK-trnT*, *atpB-rbcL*, *trnL-trnF* and *trnH-psbA* intergenic spacers and their positions are given next to the UPGMA tree.

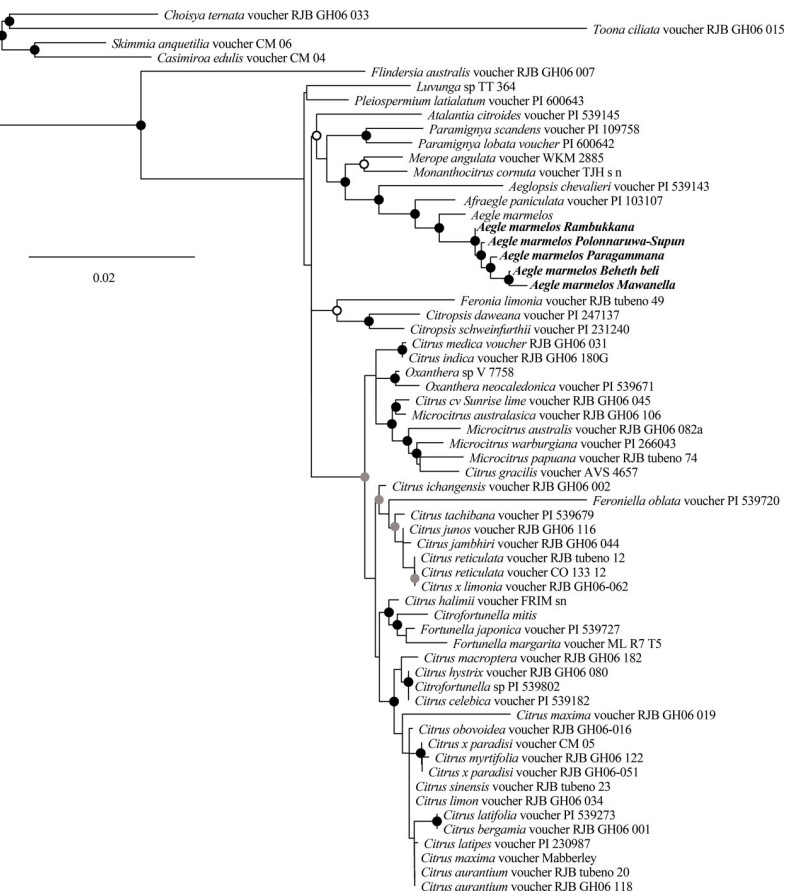

**Fig 7. The majority rule consensus tree drawn in Maximum Likelihood framework for combined markers (*matK-trnT*, *atpB-rbcL*, and *trnL-trnF* intergenic spacers).** The black dots indicate the nodes with Posterior probability (PP) >90 and bootstrap (bs) >70; the gray dots indicate the nodes with bs>70; the black circles indicate the nodes are having pp>90. The operational taxonomic units with bold letters indicate the sequences generated in the present study.

PS from the Dry Zone (Table 1). The fruit size and pulp parameters did not exhibit significant variation within accession or among the fruiting seasons of each accession (Tables 2 and 3). However, fruit size and pulp parameters were significantly different among accessions indicating higher genetic control on fruit morphometric variation than the effect of environmental factors. The higher genetic regulation of fruit size is a common phenomenon reported for various plant species such as cherry, olive, and tomato [39–41]. The variation of the fruit size and pulp parameters indicate that RA and PS are the most industrially desirable cultivars. The probable reason for selecting BB (*Beheth Beli* means medicinal bael) and PA is the preference shown by the rural people and indigenous medical practitioners towards them. The small-fruited accessions BB and PA are used in local medicinal applications regardless of the smaller fruit size and less preferred taste. The PC biplot fruit size parameters indicated that there are two size-based bael groups present within the elite accessions. However, within the two groups separately, PS and MA possess bigger fruits (Fig 2). The organoleptic assessment also revealed that RA and PS are the two best-preferred accessions with the highest preference ratings (Fig 3). The PC loading plot for weighted organoleptic scores revealed that aroma, sweetness, texture, and external appearance are more related to the overall preference than flesh color. It is

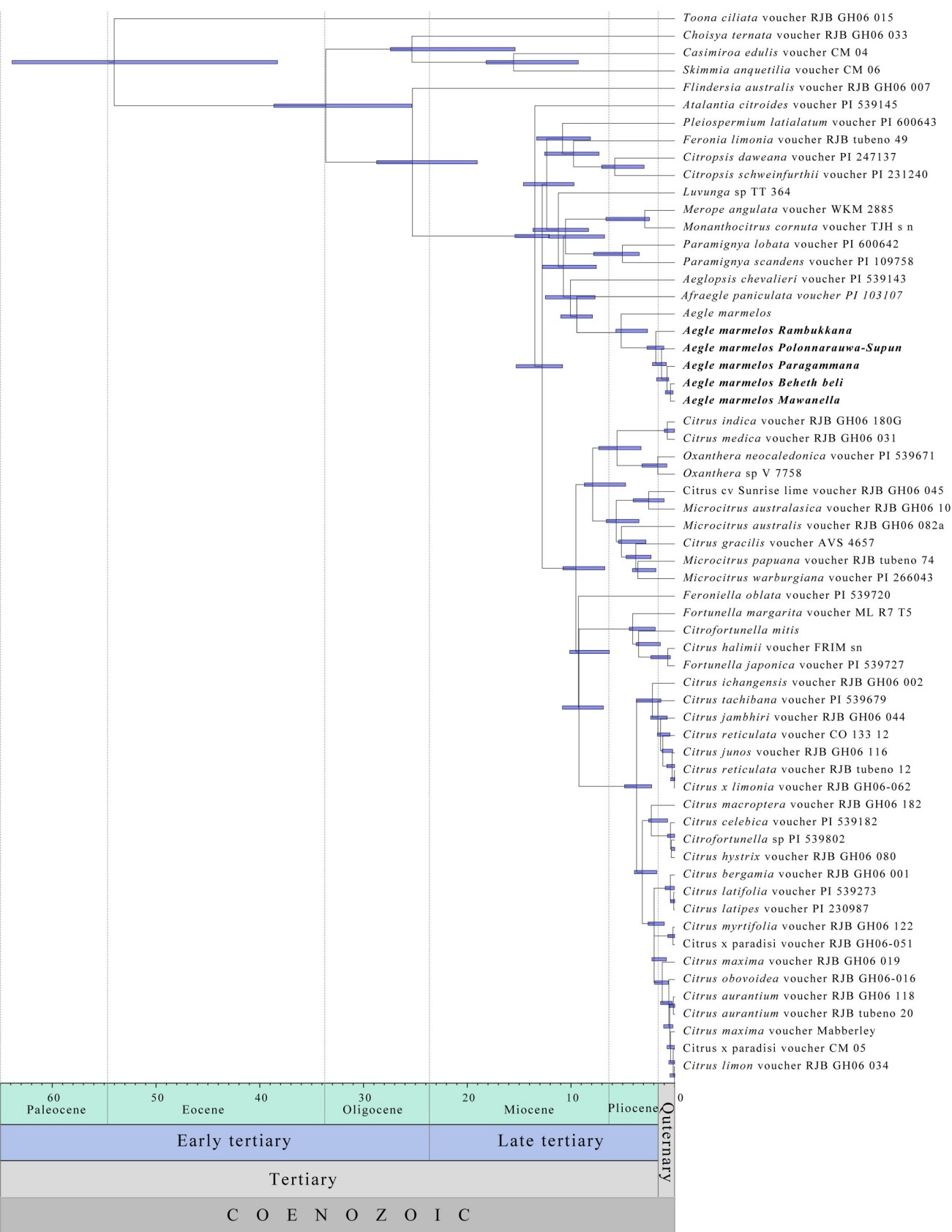

**Fig 8. The time calibrated Maximum Clade Credibility tree.** The geological time scale is given in the bottom of the diagram. The 95% highest posterior densities node bars are given at the nodes of the tree. The time scale is given below the tree.

also interesting to note that as organoleptic parameters, aroma and sweetness are highly correlated to each other (S3B Fig).

The ICP-MS elemental analysis revealed that bael does not contain sodium. The contents of Mg and K detected in bael samples were comparable with the elemental contents reported in previous studies on bael [42–44]. It is apparent from the PC loading plot for ICP-MS-based elemental contents that the elements absorbed/present as groups. The contents of isotopes, Fe, and Pb, are vert tightly correlated to each other (S4B Fig). The absorption of elements to the plants as clusters is a known mechanism in plant species [45, 46]. Further studies are needed to ascertain how the different element groups get absorbed into the plants under common regulatory mechanisms. Such studies are vital as the elemental contents within plant tissues are important as nutrients and, in some cases, as toxic substances (exceeding Lethal Dose, 50%) that can cause diseases. The PC biplots drawn for fruit size, taste, and elemental contents revealed a distinct separation of accessions in approximately similar passions (Figs 2, 4 and 5). Thus, the simultaneous selection for the bigger fruit size, better taste, and higher medicinal/ nutritional values decided by the elemental contents could have been the driving force of the domestication of bael.

According to our analysis, the divergence between two genera *Afraaegle* and *Aegle* had occurred in the early Miocene (Fig 8). The current understanding of plate tectonics where Indian plate collided with the Eurasian plate which opened the land connectivity from Africa to India in the early Miocene. This land connectivity might have opened the introduction of new plant species through animal movements. One possible hypothesis is the common ancestor of *Afraaegle* and *Aegle* could have been introduced to India and Sri Lanka in the early Miocene and evolved to be two separate lineages through isolation. The clustering of Sri Lankan and Indian bael as sister clades (Fig 7) implies that the two sources of germplasm are related to each but not quite the same. Our time analysis suggested that the movement of bael germplasm from India to Sri Lanka occurred in the Pliocene epoch. The land connectivity between Sri Lanka and India in Pliocene ice ages may have aided the germplasm movement. Soon after the Pliocene ice ages, rise of the sea levels might have acted as a robust geographic barrier that restricted possible gene flow between Sri Lankan ad Indian germplasms. As we assessed a selected set of bael accessions in Sri Lanka, it is liable to argue that the naturalization of the bael in Sri Lanka has occurred during the Quaternary period (Fig 8). Where, the slow adoption may have led to the sizeable variability in fruit morphology, taste, and elemental contents.

## Conclusions

The elite bael accessions selected for the mass propagation in Sri Lanka exhibits higher diversity in fruit size, taste, and elemental contents. The largest fruits and most preferred taste were reported for the accessions, RA, and PS. Also, on average, 1.3 of PS fruits and 2.1 RA fruits are enough to extract one kilogram of net-pulp, indicating the superior industrial value in making beverage, jam, and sweets. The bael fruit pulp does not contain sodium; however, it is apparent that the elements are absorbed/present as groups. The phylogenetic and divergence dating analyses revealed that bael got introduced to Sri Lanka in the Pliocene epoch. Our analysis also showed that *A. marmelos* is evolutionary more closely related to genus *Afraaegle* which is native to African region, showing the ancestry of *Aegle* could be from African region.

## Supporting information

**S1 Table. DNA barcoding markers and PCR conditions.** Initial Denaturation (I.D.); Denaturation (D); Primer Annealing (PA); Extension (E); Final Extension (F.E.).
(DOCX)

**S2 Table. Fruit pulp and size, organoleptic and ICP-MS based elemental data generated in the study.**
(XLSX)

**S3 Table. The DNA sequences assessed in the present study.**
(XLSX)

**S4 Table. Details of the PCA for fruit size parameters.**
(DOCX)

**S5 Table. Details of the PCA for fruit taste parameters.**
(DOCX)

**S6 Table. Details of the PCA for elemental contents.** EV: Eigen value; PV: Proportion of variance; CV: Cumulative variance.
(DOCX)

**S1 Fig. Agarose gel images of the PCR products of bael accessions for DNA barcoding markers.** A: *atpB-rbcL*; B: *trnH-psbA*; C: *matk-trnT*; D: *tRNA-leu*. L: 50 bp ladder, 1: *Beheth Beli* (BB); 2: *Paragammana* (PA); 3: *Mawanalla* (MA); 4: *Rambukkana* (RA); 5: *Polonnaruwa-Supun* (PS); +: positive control (DNA of the apple variety Spartan); -: negative control (PCR mixture without template DNA).
(TIF)

**S2 Fig. PC loading status for fruit size traits.** A: Scree plot; B: Loading plot. The contribution of the PC to the total variance is given in parenthesis.
(TIF)

**S3 Fig. PC loading status for fruit taste traits.** A: Scree plot; B: Loading plot. The contribution of the PC to the total variance is given in parenthesis.
(TIF)

**S4 Fig. PC loading status for elemental contents.** A: Scree plot; B: Loading plot. The contribution of the PC to the total variance is given in parenthesis.
(TIF)

## Acknowledgments

Department of Geology, Faculty of Science, University of Peradeniya, Sri Lanka for kindly providing the ICP-MS facilities.

## Author Contributions

**Conceptualization:** Chamila Kumari Pathirana, Terrence Madhujith, Janakie Prasanthika Eeswara, Suneth Sithumini Sooriyapathirana.

**Data curation:** Chamila Kumari Pathirana, Lahiru Thilanka Ranaweera, Janakie Prasanthika Eeswara.

**Formal analysis:** Chamila Kumari Pathirana, Lahiru Thilanka Ranaweera.

**Funding acquisition:** Chamila Kumari Pathirana, Terrence Madhujith, Janakie Prasanthika Eeswara, Suneth Sithumini Sooriyapathirana.

**Investigation:** Chamila Kumari Pathirana, Lahiru Thilanka Ranaweera.

**Methodology:** Chamila Kumari Pathirana, Lahiru Thilanka Ranaweera, Kalyani Weerasinghe Ketipearachchi, Kumar Lakshman Gamlath, Janakie Prasanthika Eeswara, Suneth Sithumini Sooriyapathirana.

**Project administration:** Terrence Madhujith, Janakie Prasanthika Eeswara, Suneth Sithumini Sooriyapathirana.

**Resources:** Chamila Kumari Pathirana, Lahiru Thilanka Ranaweera, Terrence Madhujith, Kalyani Weerasinghe Ketipearachchi, Kumar Lakshman Gamlath, Janakie Prasanthika Eeswara, Suneth Sithumini Sooriyapathirana.

**Software:** Chamila Kumari Pathirana, Lahiru Thilanka Ranaweera.

**Supervision:** Janakie Prasanthika Eeswara, Suneth Sithumini Sooriyapathirana.

**Validation:** Chamila Kumari Pathirana, Terrence Madhujith, Janakie Prasanthika Eeswara, Suneth Sithumini Sooriyapathirana.

**Visualization:** Chamila Kumari Pathirana, Lahiru Thilanka Ranaweera, Terrence Madhujith, Janakie Prasanthika Eeswara, Suneth Sithumini Sooriyapathirana.

**Writing – original draft:** Chamila Kumari Pathirana, Lahiru Thilanka Ranaweera, Terrence Madhujith, Janakie Prasanthika Eeswara.

**Writing – review & editing:** Chamila Kumari Pathirana, Lahiru Thilanka Ranaweera, Terrence Madhujith, Janakie Prasanthika Eeswara.

**Writing - Original Draft:** Suneth Sithumini Sooriyapathirana.

**Writing - Review & Editing:** Suneth Sithumini Sooriyapathirana.

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
