## [Decision Letter · Decision Letter 0]

27 Mar 2020

PONE-D-19-29210

Assessment of the elite accessions of bael [Aegle marmelos (L.) Corr.] in Sri Lanka based on morphometric, organoleptic, and elemental properties of the fruits and phylogenetic relationships

PLOS ONE

Dear Dr. Sooriyapathirana,

Thank you for submitting your manuscript to PLOS ONE. After careful consideration, we feel that it has merit but does not fully meet PLOS ONE’s publication criteria as it currently stands. Therefore, we invite you to submit a revised version of the manuscript that addresses the points raised during the review process.

We would appreciate receiving your revised manuscript by May 1. To enhance the reproducibility of your results, we recommend that if applicable you deposit your laboratory protocols in protocols.io, where a protocol can be assigned its own identifier (DOI) such that it can be cited independently in the future. For instructions see: http://journals.plos.org/plosone/s/submission-guidelines#loc-laboratory-protocols

We look forward to receiving your revised manuscript.

Kind regards,

Zhenhai Han, PhD

Academic Editor

PLOS ONE

Journal Requirements:

2. In your Methods, please confirm that the members of the organoleptic panel provided informed consent for their participation in the study, and state whether consent was written or oral.

Reviewers' comments:

Reviewer's Responses to Questions

**Comments to the Author**

1. Is the manuscript technically sound, and do the data support the conclusions?

Reviewer #1: Yes

2. Has the statistical analysis been performed appropriately and rigorously? 

Reviewer #1: Yes

3. Have the authors made all data underlying the findings in their manuscript fully available?

Reviewer #1: Yes

4. Is the manuscript presented in an intelligible fashion and written in standard English?

Reviewer #1: Yes

5. Review Comments to the Author

Reviewer #1: In my opinion, bael is an important plant, but not characterized scientifically. So, this paper is important. However, I have several concerns before publication.

Major concerns

1. The description of bael may be incorrect. As I live in Japan, I am not familiar with this plant. But I found this plant in southeast Asian countries such as Thailand. The description in Wikipedia supports my observation. Although the tree is sacred by Hindus, I found these plants in some Buddhist temples in Bhutan and Thailand. So, please gather the detailed information on this plant and reflect them on the paper.

2. In recent years, the phylogenetic study of Aurantioideae has advanced. Mr. Ted Cole and Dr. Milton Groppo summarized the current progress. So, search the website using the term "Rutaceae Phylogeny Poster." Analysis of the papers cited in the poster is important. The authors included Aeglopsis in the current phylogenetic analysis but did not include Afraegle, an African plant. Afraegle is more genetically similar to Aegle. The divergent time between these two may be similar to that of multiple species of Citrus (probably 10MYA or below). It may be similar to the divergent time of Indian bael and Sri Lankan bael. So, using the available data, please compare the Aegle sequences with Afraegle sequences. Additional experiment is not required.

3. The authors concluded that the Indian plant and Sri Lankan plant split 8.52 MYA. I agree with the result of this calculation, but disagree with the authors' suggestion "bael got introduced from India to Sri Lanka 8.52 MYA in the Pliocene epoch." The genetic diversity of bael in India may be wide, and some of the Indian plants may be genetically similar to Sri Lankan plants. Because the data of the Indian plant in this paper is from only one plant, the authors did not exclude this possibility.

Minor concerns

1. Please show the contribution ratios in every figure for PCA.

2. S1 Fig. Fragment lengths of PCR products are different between positive control and your samples. Why? Information on the DNA marker is absent.

3. Accession numbers of the sequence data are absent.

4. Line 170: "Takara" is "Takara Bio."

5. PCA (Fig.5): PC3 or above may contain important information. So, as the supplementary figure, please make a figure containing additional components.

6. PLOS authors have the option to publish the peer review history of their article (what does this mean?). If published, this will include your full peer review and any attached files.

Reviewer #1: Yes: Yukio Nagano

---

## [Author Response · Author response to Decision Letter 0]

17 Apr 2020

April 17, 2020

Academic Editor,

PLoS ONE

Submission of the Revised Manuscript [PONE-D-19-29210] 

On behalf of all the authors, I wish to pay my sincere thanks for taking necessary actions to review the manuscript on ‘Assessment of the elite accessions of bael [Aegle marmelos (L.) Corr.] in Sri Lanka based on morphometric, organoleptic, and elemental properties of the fruits and phylogenetic relationships’. Here with I am submitting the revised article after incorporating the suggestions given by reviewers. The answers and our actions for each question/comment/suggestion are given below.

Comment One: Thank you for submitting your manuscript to PLOS ONE. After careful consideration, we feel that it has merit but does not fully meet PLOS ONE’s publication criteria as it currently stands. Therefore, we invite you to submit a revised version of the manuscript that addresses the points raised during the review process.

Answer: Thank you very much for the appreciation and the invitation to submit the revised version. We are really honored and humbled by this decision.

Recommendation 1: Please ensure that your manuscript meets PLOS ONE's style requirements, including those for file naming. The PLOS ONE style templates can be found at http://www.plosone.org/attachments/PLOSOne_formatting_sample_main_body.pdf and http://www.plosone.org/attachments/PLOSOne_formatting_sample_title_authors_affiliations.pdf

Answer: Thank you! The manuscript was edited to meet the PloS ONE guidelines as given in the two templates.

Recommendation 2: In your Methods, please confirm that the members of the organoleptic panel provided informed consent for their participation in the study, and state whether consent was written or oral.

Answer: The written informed consent was obtained from each participant of the organoleptic panel. A statement is added to the relavent section of the Methods. 

Recommendation 3: We suggest you thoroughly copyedit your manuscript for language usage, spelling, and grammar. If you do not know anyone who can help you do this, you may wish to consider employing a professional scientific editing service. Upon resubmission, please provide the following:

Answer: Thank you very much for recommending the language corrections. The entire manuscript is thoroughly copyedited for language usage, spelling, and grammar using Grammarly software (https://app.grammarly.com/).

Recommendation 4: We note that you have indicated that data from this study are available upon request. PLOS only allows data to be available upon request if there are legal or ethical restrictions on sharing data publicly. For more information on unacceptable data access restrictions, please see http://journals.plos.org/plosone/s/data-availability#loc-unacceptable-data-access-restrictions.

Answer: Thank you! We wish to change the Data Availability Statement as we don’t have any restrictions to make the data publicly available. The DNA sequence data generated in the present study are available in https://www.ncbi.nlm.nih.gov/nuccore/ under the accession numbers MN082783 to MN082802. The other datasets presented in this study are provided either in the manuscript or supporting information. A Supplementary Table (S2 Table) was prepared to showcase the raw data fruit size, taste and pulp properties and ICP-MS based elemental contents (an Excel file with named worksheets).

Reviewer's Responses to Questions

Comments to the Author

1. Is the manuscript technically sound, and do the data support the conclusions?

Reviewer #1: Yes

Answer: Thank you for the recognition.

2. Has the statistical analysis been performed appropriately and rigorously?

Reviewer #1: Yes

Answer: Thank you!

3. Have the authors made all data underlying the findings in their manuscript fully available?

Reviewer #1: Yes

Answer: Thank you!

4. Is the manuscript presented in an intelligible fashion and written in standard English?

Reviewer #1: Yes

Answer: Thank you!

5. Review Comments to the Author

Reviewer #1: In my opinion, bael is an important plant, but not characterized scientifically. So, this paper is important. However, I have several concerns before publication.

Answer: Thank you! In the revised manuscript, all the concerns were addressed and the sections were modified accordingly.

Major concerns

1. The description of bael may be incorrect. As I live in Japan, I am not familiar with this plant. But I found this plant in southeast Asian countries such as Thailand. The description in Wikipedia supports my observation. Although the tree is sacred by Hindus, I found these plants in some Buddhist temples in Bhutan and Thailand. So, please gather the detailed information on this plant and reflect them on the paper.

Answer: The recommended information was collected and the manuscript was modified accordingly.

2. In recent years, the phylogenetic study of Aurantioideae has advanced. Mr. Ted Cole and Dr. Milton Groppo summarized the current progress. So, search the website using the term "Rutaceae Phylogeny Poster." Analysis of the papers cited in the poster is important. The authors included Aeglopsis in the current phylogenetic analysis but did not include Afraegle, an African plant. Afraegle is more genetically similar to Aegle. The divergent time between these two may be similar to that of multiple species of Citrus (probably 10MYA or below). It may be similar to the divergent time of Indian bael and Sri Lankan bael. So, using the available data, please compare the Aegle sequences with Afraegle sequences. Additional experiment is not required.

Answer: We appreciate this valuble suggestion to look at similar genera to Aegle especially the ones that diverged out to Africa. However, there is one Afaegle species (Afraegle paniculata) is only genetically described for the markers we have used. In the present study, we used four makers (matK-trnH, atpB-rbcL, trnL-trnF, and trnH-psbA) to carry out the phylogenetic analysis. When we searched for all the sequence data available for Afraegle paniculata for the markers, we found following GenBank Accession numbers, matK-trnH - EF138838, atpB-rbcL - EF126501, trnL-trnF - AY295295 and no sequence was found for trnH-psbA. In the poster published by Mr. Ted Cole and Dr. Milton, a list of references given. However, only above three sequences can be found for Afraaegle paniculata. We reanlysed the sequence data for phylogenetic relationships with Afraaegle paniculata sequences and developed new Figures for phylogenetic tree (Fig 7) and time tree (Fig 8). The sentences were added to explain the new results to the Sections of Results, and Discussion. 

3. The authors concluded that the Indian plant and Sri Lankan plant split 8.52 MYA. I agree with the result of this calculation, but disagree with the authors' suggestion "bael got introduced from India to Sri Lanka 8.52 MYA in the Pliocene epoch." The genetic diversity of bael in India may be wide, and some of the Indian plants may be genetically similar to Sri Lankan plants. Because the data of the Indian plant in this paper is from only one plant, the authors did not exclude this possibility.

Answer: Thank you very much for your expert insight regarding this estimation. It is true that we cannot rule out the possibility of having similar bael genetic diversity in Sri Lanka to that of India as we did not find multiple samples from India. We understand that more studies are needed to prove this hypothetical inference. Therefore the relavent sections in the paper were modified avoid this statement.

Minor concerns

1. Please show the contribution ratios in every figure for PCA.

Answer: All the PCA Figures were modified to include the PC contributions.

2. S1 Fig. Fragment lengths of PCR products are different between positive control and your samples. Why? Information on the DNA marker is absent.

Answer: We really appreciate this query on DNA band sizes. As the positive control we used the DNA of the apple variety Spartan because it was very easy to amplify in PCR. In trnH-psbA, we found a big size difference between bael bands and the apple band. Pang et al., (2012), indicated that trH-psbA amplicon sizes vary greatly from 152 bp to 1006 bp. This might be the reason for our observation and we have observed these kinds of big difference among other plant species as well.

Pang X, Liu C, Shi L, Liu R, Liang D, Li H, et al. 2012 Utility of the trnH–psbA intergenic spacer region and its combinations as plant DNA barcodes: a meta-analysis. PLoS ONE. 2012: 7(11):e48833. https:// doi.org/10.1371/journal.pone.0048833.

The ladder type (50 bp) was included in the S1 Figure caption. Also, with this suggestion, we included the key band sizes of the ladder in gel images.

3. Accession numbers of the sequence data are absent.

Answer: The Sequence data are available in https://www.ncbi.nlm.nih.gov/nuccore/ under the accession numbers MN082783 to MN082802. In the S3 Table, these numbers are displayed with the other popset subjected to analyses

4. Line 170: "Takara" is "Takara Bio."

Answer: Modified as recommended.

5. PCA (Fig.5): PC3 or above may contain important information. So, as the supplementary figure, please make a figure containing additional components.

Answer: Thank you for this suggestion. In S4C Figure, a scatter plot was drwan using PC1, PC2, and PC3. 

Again, on behalf of all the authors, I highly appreciate your revisions and editorial remarks to improve the manuscript. We look forward to the next step of the publication process.

Thanking you!

Sincerely,

…………………………………..

Prof. S.D.S.S. Sooriyapathirana 

Corresponding Author

Email: sunethuop@gmail.com

---

## [Decision Letter · Decision Letter 1]

11 May 2020

Assessment of the elite accessions of bael [Aegle marmelos (L.) Corr.] in Sri Lanka based on morphometric, organoleptic, and elemental properties of the fruits and phylogenetic relationships

PONE-D-19-29210R1

Dear Dr. Sooriyapathirana,

We are pleased to inform you that your manuscript has been judged scientifically suitable for publication and will be formally accepted for publication once it complies with all outstanding technical requirements.

With kind regards,

Zhenhai Han, PhD

Academic Editor

PLOS ONE

Additional Editor Comments (optional):

Reviewers' comments:

Reviewer's Responses to Questions

**Comments to the Author**

1. If the authors have adequately addressed your comments raised in a previous round of review and you feel that this manuscript is now acceptable for publication, you may indicate that here to bypass the “Comments to the Author” section, enter your conflict of interest statement in the “Confidential to Editor” section, and submit your "Accept" recommendation.

Reviewer #1: (No Response)

2. Is the manuscript technically sound, and do the data support the conclusions?

Reviewer #1: (No Response)

3. Has the statistical analysis been performed appropriately and rigorously? 

Reviewer #1: (No Response)

4. Have the authors made all data underlying the findings in their manuscript fully available?

Reviewer #1: (No Response)

5. Is the manuscript presented in an intelligible fashion and written in standard English?

Reviewer #1: (No Response)

6. Review Comments to the Author

Reviewer #1: (No Response)

7. PLOS authors have the option to publish the peer review history of their article (what does this mean?). If published, this will include your full peer review and any attached files.

Reviewer #1: Yes: Yukio Nagano

---

## [Editor Report · Acceptance letter]

14 May 2020

PONE-D-19-29210R1 

Assessment of the elite accessions of bael [*Aegle marmelos* (L.) Corr.] in Sri Lanka based on morphometric, organoleptic, and elemental properties of the fruits and phylogenetic relationships 

Dear Dr. Sooriyapathirana:

I am pleased to inform you that your manuscript has been deemed suitable for publication in PLOS ONE. Congratulations! Your manuscript is now with our production department. 

With kind regards,

on behalf of

Dr. Zhenhai Han 

Academic Editor

PLOS ONE